# Exploring the Effects of Brain Stimulation on Musical Taste: tDCS on the Left Dorso-Lateral Prefrontal Cortex—A Null Result

**DOI:** 10.3390/brainsci12040467

**Published:** 2022-03-31

**Authors:** Gemma Massetti, Carlotta Lega, Zaira Cattaneo, Alberto Gallace, Giuseppe Vallar

**Affiliations:** 1School of Medicine and Surgery, University of Milano-Bicocca, 21026 Milan, Italy; 2Milan Center for Neuroscience (NeuroMI), 20126 Milan, Italy; carlotta.lega@unimib.it (C.L.); giuseppe.vallar@unimib.it (G.V.); 3Mind and Behavior Technological Center, 20126 Milan, Italy; 4Department of Psychology, University of Milano-Bicocca, 20126 Milan, Italy; 5Department of Human and Social Sciences, University of Bergamo, 24129 Bergamo, Italy; zaira.cattaneo@unibg.it; 6IRCCS Mondino Foundation, 27100 Pavia, Italy

**Keywords:** neuroesthetics, music, liking, emotions, tDCS, DLPFC

## Abstract

Humans are the only species capable of experiencing pleasure from esthetic stimuli, such as art and music. Neuroimaging evidence suggests that the left dorsolateral prefrontal cortex (DLPFC) plays a critical role in esthetic judgments, both in music and in visual art. In the last decade, non-invasive brain stimulation (NIBS) has been increasingly employed to shed light on the causal role of different brain regions contributing to esthetic appreciation. In Experiment #1, musician (*N* = 20) and non-musician (*N* = 20) participants were required to judge musical stimuli in terms of “liking” and “emotions”. No significant differences between groups were found, although musicians were slower than non-musicians in both tasks, likely indicating a more analytic judgment, due to musical expertise. Experiment #2 investigated the putative causal role of the left dorsolateral pre-frontal cortex (DLPFC) in the esthetic appreciation of music, by means of transcranial direct current stimulation (tDCS). Unlike previous findings in visual art, no significant effects of tDCS were found, suggesting that stimulating the left DLPFC is not enough to affect the esthetic appreciation of music, although this conclusion is based on negative evidence,.

## 1. Introduction

“Neuroesthetics” aims to elucidate the neural bases and mechanisms underlying “liking” and related emotions, using art as a means to understand these functions of the human brain [1,2,3,4]. In the last decade, an increasing number of studies has been devoted to the analysis of esthetic experience, which started to be an interdisciplinary science, rather than being confined to visual art. Among the various disciplines related to art, music has been receiving increasing interest [5,6]. Over the years, various studies have been carried out concerning the putative differences between musicians and non-musicians. Nevertheless, this line of research focused mostly on plastic changes, such as the outcome of continuous and prolonged engagement with music, and the cognitive domains involved in the perception and elaboration of musical stimuli, initially suggesting that music processing involves regions of the right hemisphere in non-musicians, and of the left hemisphere in musicians [7,8,9,10]. A recent review of Criscuolo and colleagues [11] highlights that music experts, as compared to laypersons, possess highly developed auditory, sensorimotor and subcortical regions (i.e., a higher volume/activity in musicians vs. non-musicians, according to an ALE meta-analysis), featuring bilateral widespread neuro-anatomical and functional differences. These findings challenge a previous and more simplistic perspective, which was focused on the left-sided lateralization of brain functions related to music, with the cognitive elaboration of a musical piece being compared to the comprehension of language [12,13,14]. In this respect, Schellenberg and Weiss [15] conclude that “musical aptitude” (i.e., natural music abilities) is associated with linguistic abilities, including phonological processing, facility in the acquisition of a second language, and, at least in part, reading [16,17]. Musical training, by contrast, would be associated with enhanced performance in a variety of listening tasks, musical or otherwise [18]. Furthermore, in childhood, musical training appears to be a predictor of good performance across a wide variety of cognitive abilities, including memory, language, and visuospatial skills [19,20,21]. However, a note of caution is suggested by the finding that the comparison between adult musicians and naïve participants often yields null results when outcome measures do not involve music or listening [22]. Gaser and Schlaug [23] suggest that differences between musicians and non-musicians are associated with plastic neural changes, due to continual rehearsal and extended skill acquisition, which lead in turn to changes in the patterns of neural activations between musicians and naïve participants in tasks involving music. A few studies investigated in more detail the putative differences between musicians and naïve participants, related to their emotional and esthetic experience while listening to music. “Esthetic experience” is defined by Brattico and Pearce [5] as “[an experience] in which the individual immerses herself/himself in the music, dedicating her attention to perceptual, cognitive, and affective interpretation based on the formal properties of the perceptual experience”. Possibly related to esthetic experience, music listening causes affective and physiological reactions, such as thrills, awe, and the state of being moved [24,25]. Even though there is evidence suggesting that musical training has very little effect on the perception of emotion in music [25] and that evaluations of music also rely on cognitive biases and heuristics that do not depend on the stimuli themselves [26], musicians and non-musicians react differently to sadness and fear, both at behavioral and physiological levels [27]. More precisely, sadness and fear conveyed by music would be more arousing for musicians, suggesting an association between musical training and the processing of “negative” emotions, while non-specific activations would be brought about by happiness. A somewhat unexpected finding from an esthetic perspective is that the lack of musical experience brings about more sensitivity to music [28,29]. In addition, musicians and non-musicians showed remarkable differences in their esthetic responses to the stimulus, in this case a piece of country music [29]. Particularly, non-musicians reported a stronger reaction to the stimuli. These differences may not involve only specific music genres but should be considered in a wider frame: an ERP study of responses to chord sequences, in which the final chord varied in terms of its harmonic congruency with the context, highlighted enhanced emotion-related neural processing for esthetic judgments, as compared with judgments about how congruent each chord sequence sounded in non-musicians but not in musicians [30]. Therefore, it may be possible that musicians rely less on the emotions and more on other (cognitive?) factors in the decision-making process about esthetic judgments on music [31]. At a behavioral level, subjective criteria for esthetic judgment of music were compared between psychology vs. music students, showing differences in their relative weighting of different criteria: in particular, psychology students tended to rate extrinsic choice criteria (e.g., “suits specific activities”) as opposed to intrinsic criteria (e.g., “interesting structure”) higher than did music students [32]. On the other hand, at a neurofunctional level, an fMRI study [33] found functional differences in the limbic system, associated with musical expertise, with musicians showing enhanced liking-related activity in the fronto-insular and cingulate cortices, as compared to laypersons, when asked to express an emotional or esthetic judgment about music. Based on these previous findings, to further investigate differences between musicians and non-musicians, Experiment #1 focused on the liking and emotional appraisal induced by musical pieces, using the same musical excerpts. Furthermore, the study aimed at setting up a task that could be useful to investigate liking and emotions in music, both in expert and in naïve participants. Currently available tools are mostly questionnaires or categorization tasks, in which participants assign musical tracks to a limited range of emotions or express a preference for specific music genres [34,35,36].

Across the years, research in the field of esthetic appreciation moved from a behavioral to a neurofunctional level of investigation. As for its neural correlates, studies investigating the neuroscience of art assign an important role to the left dorsolateral prefrontal cortex (DLFPC) [1,4,37]. The left DLFPC may be also involved in the esthetic experience of music, possibly in relationship to its role in the reward circuit [38,39]. Music might evoke a large variety of emotions that, independent of their valence, may be associated with some sort of pleasure [40,41]. Huron [42] suggested that those feelings may arise from the structural and temporal features of music, such as anticipation and expectancy. As for the neural underpinnings of these behavioral findings, the evidence from neuroimaging studies concerned with music has led to a model in which perceptual expectations and predictions generated in the auditory cortex may acquire affective value through the engagement of fronto-striatal circuits, which involve the ventral and the dorsal subdivisions of the striatum (especially, the *nucleus accumbens* (NAcc) and the right caudate nucleus) [43,44]. According to this model, the experience of a music reward would be driven by the functional link between the auditory perceptual/cognitive mechanisms, on the one hand, and the evaluative/reward mechanisms, on the other hand. Perceived pleasure, psychophysiological measures of emotional arousal, and the monetary value assigned to music, are all significantly increased by exciting fronto-striatal pathways, particularly the left DLPFC, via transcranial magnetic stimulation (TMS). Conversely, the inhibition of this system leads to decreases in all these variables, as compared with sham stimulation [6]. Changes in activity in the NAcc are associated with the bidirectional modulation of both hedonic and motivational behavioral responses. The TMS-induced changes in the fMRI-assessed functional connectivity between the NAcc and the frontal and auditory cortices predict the degree of modulation of hedonic responses [39]. In sum, the left DLPFC appears to play an important role in the communication between regions involved in the esthetic processes, possibly due to its role in dopamine releasing and in blood-oxygen-level-dependent (BOLD) activity regulation in reward-related structures [45,46]. Experiment #2 aimed at investigating the role of the left DLPFC in the esthetic liking of musical excerpts through transcranial direct current stimulation (tDCS), which exerts a modulatory effect on the cerebral cortex altering excitability and activity, which is dependent on the current flow direction through the target neurons [47]. In fact, consistent evidence suggests that anodal tDCS (i.e., anode placed over the region of interest) causes an enhancement of cortical excitability during stimulation, which lasts several minutes after the end of the stimulation and is usually accompanied by an enhancement of cognitive performance (e.g., attention, executive functions, memory, etc.) [48]. Overall, the present research aimed at investigating the esthetic judgment in music: firstly at a behavioral level, comparing esthetic evaluation and emotional valence in professional musicians and laypersons; secondly, at a neurofunctional level, probing the role of the DLPFC through tDCS. The study aimed then at assessing the possible role of regions involved in the neuroesthetics of art [1], in that of music [38,39]. 

## 2. Materials and Methods

### 2.1. Experiment 1

#### 2.1.1. Participants

Forty participants, divided in two groups, volunteered to participate in this study. The non-musician group included 20 students from the University of Milano-Bicocca (F = 10; mean age = 22.55 years-old, S.D. 2.24; range 19–26) who reported no previous training or special interest in music (e.g., music bloggers). The musician group included 20 musicians (F = 10; mean age = 21.5, S.D. 3.26; range 18–31 years old) with musical education (graduating or graduated students of the Italian Musical Conservatory). Exclusion criteria were uncorrected hearing or visual impairments. All participants were right-handed [49] and had normal hearing. Written informed consent was obtained from all participants. The local ethical committee of the University of Milano-Bicocca approved the experiment and participants were treated in accordance with the Declaration of Helsinki.

#### 2.1.2. Stimuli

Stimuli consisted of 45 excerpts cut out from musical pieces of different cultures and different musical genres. None of them had been used for soundtracks of movies or television spots. The excerpts never corresponded to the beginning of the track or containing vocal parts, to rule out the possibility of recognizing the track, or of introducing other confounding variables, due to the presence of language. Excerpts performed in various formations (soloist, duo, trio, quartet, ensemble, and orchestra) were selected; a third of them had a tonal structure, a third was, instead, atonal and the remaining tracks had a mixed structure. The instruments used were both canonical traditional (e.g., violin, guitar, piano, flute, etc.) and electronic (e.g., electric guitar, electronic keyboard, electronic drums, etc.). Furthermore, stimuli that did not contain a melodic line, but had just a rhythmic structure, were also included. The tracks were cut, so that each of them lasted 7 s, and, subsequently, were equalized for intensity. Finally, a fade-out effect in the last 1500 msec was applied. All adjustments were made through the Audacity© software (https://www.audacityteam.org/, accessed on 1 February 2020).

As for the choice of the excerpts, the selected tracks were as balanced as possible in terms of variability of rhythm, melody, speed, dynamics, density, structure, and instruments. Excerpts were then classified in three categories (15 tracks each): “beautiful”, “ugly”, “unsure judgment” (“medium” level, neither “beautiful” nor “ugly”) through a pilot study in which five western professional musicians, graduated at the Conservatory, were enrolled, and asked to classify the tracks into the three categories proposed, according to their professional judgment.

As a result, the “beautiful” category was mostly composed of excerpts with a tonal framework, a regular rhythmic structure, or easily perceptible and consonant sounds. On the contrary, the “ugly” category was mostly composed of excerpts with an atonal framework, an irregular rhythmic structure, or hardly perceptible and dissonant sounds. The “medium” category included excerpts with mixed characteristics, as compared to the other two groups. 

#### 2.1.3. Procedure

Participants, seated in front of a 15.500 PC (1280 × 800 pixels) screen at an approximate distance of 57 cm, in a normal-lightened and silent room, were asked to perform a computerized rating task. Before starting the experiment, participants were informed that they would be performing two computer tasks, in which they would be listening to a few short musical excerpts through earphones (sound-proof over-ear headphones). The first task required an esthetic judgment (“How much do you like this excerpt?”); while the second task required an expression of the emotional valence evoked by the excerpt (“Positive or negative emotion?”). Figure 1 shows the timeline of the two experimental trials. Participants listened to the excerpt of seven seconds, laps of time in which the display of the computer was completely white. After that, a vertical bar appeared in the center of the screen. In order to make the judgment easier, the bar was filled with a gradient of colors from bottom to top, from red to green. Participants were instructed to express their judgment (esthetic or emotional, depending on the task) by clicking with the mouse using their right hand. The mouse cursor was a fully horizontal arrow that appeared on the right upper extreme of the bar and moved only vertically. Participants were informed that the bar was meant to express a 0–100% scale: the lower end of the bar (red) corresponded to a zero level of liking (or a very negative emotion, depending on the task), whereas the upper end of the bar (green) corresponded to the maximum level of liking (or a very positive emotion). The bar remained visible until participants expressed their judgment. Then, after 300 ms, a new track was played. There was no time limit, but participants were encouraged to respond as fast and accurately as possible. The presentation order of the tasks (esthetic appreciation and emotional judgment) was counterbalanced across participants and within male and female participants. Excerpts were presented in random order. The whole experimental session lasted approximately 45 min, with a brief pause between the two tasks. The software E-prime 2.0 (Psychology Software Tools, Inc., Pittsburgh, PA, USA) was used for stimuli presentation and data recording.

#### 2.1.4. Data Analysis

The position of the mouse cursor along the bar was automatically converted by the software to percentage rating scores, where a 0% score corresponded to the mouse cursor positioned at the lower end (red) of the rating bar and a 100% score corresponded to the mouse cursor positioned at the upper end (green) of the rating bar. Analyses were run using SPSS© 19. Bayesian analyses were performed using the JASP software (Version 0.16.0) (https://jasp-stats.org, accessed on 1 March 2021). A Bayesian ANOVA with an analysis of the effects [44] was performed. Such an analysis estimates the inclusion Bayes factor (BF_incl_), which can be interpreted as evidence in the data for including a predictor, either a main effect or an interaction. BFs_incl_ between 1 and 3, 3 and 10, and larger than 10, are considered respectively as anecdotal, moderate, and strong evidence for including a predictor; conversely, BFs_incl_ between 1 and 1/3, 1/3 and 1/10, and smaller than 1/10 indicate anecdotal, moderate, or strong evidence for excluding a predictor. A BF_incl_ = 1 indicates no evidence in favor of including or excluding a predictor [50]. Bayesian post hoc tests, based on the default t-test with a Cauchy (0, r = 1/sqrt(2)) prior, were performed to further interpret the results of the best-fitting model. The resulting Bayes Factors (BF_10_) compare the alternative hypothesis that two levels of a factor differ from each other with the null hypothesis of no difference, the thresholds for supporting the alternative vs. null hypotheses being the same as the one illustrated above for including or excluding a factor using BF_incl_.

### 2.2. Experiment 2

#### 2.2.1. Participants

Twenty-two university students (F = 16; mean age ± SD = 22 ± 1.91 years-old; range 20–27 years-old), divided in two groups, who reported no previous training or special interest in music volunteered to participate in this study (e.g., music bloggers). None of them had taken part in Experiment #1. Furthermore, as no significant differences were found between musicians and laypersons neither in the esthetical nor in the emotional judgment, we decided not to include a second group of professional musicians. Participants were tested all right-handed [43] and had normal hearing. All participants had no history or evidence of any neurologic, psychiatric, or other medical disease. Specifically, participants had none of the following: family history of epilepsy, current pregnancy, cardiac pacemaker, previous surgery involving implants to the head (cochlear implants, aneurysm clips, or brain electrodes). Finally, the participants did not take any medication. Participants were randomly divided in two groups, each including 11 subjects (first group: F = 9; mean age ± SD = 21.55 ± 1.69 years-old, range 20–25; second group: F = 7; mean age ± SD = 22.45 ± 2.16 years-old, range 20–27). Written informed consent was obtained from all participants. The ethical committee of the University of Milano-Bicocca approved the experiment and participants were treated in accordance with the Declaration of Helsinki.

#### 2.2.2. Stimuli

Two different sets of musical excerpts were set up, each including 42 musical excerpts. Set 1 was the same set used in Experiment #1. In the esthetic judgment task, three out of 45 tracks, which had obtained, on average, the most extreme judgment scores, were excluded, to adopt only the excerpts with a certain degree of modulation in their esthetic judgment. Set #2, instead, was created ad hoc for this experiment. Within the same songs the tracks used for Set #1, new excerpts, alike to the ones chosen for Set #1, with reference to the criteria of harmonic, rhythmic and melodic structure, were selected, to obtain two largely superimposable, but not identical, sets. This was verified through independent samples *t*-tested between Group #1 and Group #2 on the pre-tDCS stimulation (to ensure no conditioning on judgments) scores obtained by participants in Set #1 vs. Set #2. No differences were found between each category of stimuli (all *p* > 0.05).

#### 2.2.3. Procedure

Participants were seated in front of a 15.500 PC (1280 × 800 pixels) screen at an approximate distance of 57 cm, in a lightened and silent room, and were asked to perform the computerized rating task of Experiment #1. Both groups performed the same task, but with a reversed presentation order of the two sets of stimuli: (a) Group #1 listened to set #2, pre-stimulation and to set #1, post-stimulation; (b) Group #2 listened to set #1, pre-stimulation and to set #2, post-stimulation.

Before starting the experiment, participants were informed that they would be listening a few short musical excerpts through earphones (sound-proof over-ear headphones). The task required an esthetic judgment (“How much do you like this excerpt?”, see Figure 1). Participants listened the excerpt of seven seconds, laps of time in which the display of the computer was completely white. After that, a vertical bar appeared in the center of the screen. To make the judgment easier, the bar was filled with a gradient of colors from the bottom to the top, from red to green. Participants were instructed to express their judgment by clicking with the mouse using their right hand. The mouse cursor was a fully horizontal arrow that appeared on the right upper extreme of the bar, and moved only vertically. Participants were informed that the bar was meant to express a 0–100% scale: the lower end of the bar corresponded to a zero level of liking, the upper end of the bar to the maximum level of liking. The bar remained visible until participants had expressed their judgment. Then, after 300 ms, a new track was played. There was no time limit for responding, but participants were encouraged to respond in the fastest and most accurate way. Excerpts were presented in randomized order. The software E-prime 2.0 (Psychology Software Tools, Inc., Pittsburgh, PA, USA) was used for stimuli presentation and data recording.

Transcranial direct current stimulation (tDCS) was delivered by a battery driven, constant current stimulator (Eldith, Neuroconn, Ilmenau, Germany) through a pair of saline-soaked sponge electrodes (7 × 5 cm: 35 cm^2^) kept firm by elastic bands. The excitability-enhancing anodal electrode was placed over the left DLPFC localized as the middle point between F3 and F5 in the 10–20 electroencephalography (EEG) system [1,51], whereas the cathodal electrode was placed over the right supra-orbitary region. This electrode arrangement (anodal electrode over one DLPFC with the cathodal electrode over the contralateral supraorbital area) is thought to induce unilateral modulation of one DLPFC and has been demonstrated to be effective in several studies [52,53,54]. Each participant underwent two stimulation sessions: real and sham. In each session, participants performed the task twice: once before stimulation, and once after stimulation. The evaluated excerpts were different in the pre- and post-tDCS assessments (see above). Sessions were separated by an average of 3.5 days (SD 1.5 days, range: 2–5 days). The order of the stimulation sessions was counterbalanced across participants, so that half started with the sham session and the other half with the real session. In the real tDCS session, stimulation intensity was set at 2 mA, and the duration of stimulation was 20 min. Previous studies have demonstrated that this intensity of stimulation is safe and can be more effective than a 1 mA stimulation [55]. Moreover, 20 min of 2 mA anodal stimulation results in an excitability enhancement that is still observable 90 min after the end of the stimulation [1]. For the sham stimulation, the electrodes were placed at the same positions as for active stimulation, but the stimulator was turned on only for 30 s. Thus, participants felt the initial itching sensation associated with tDCS, but received no active current for the rest of the stimulation period. This procedure ensured that participants felt the initial itching sensation at the beginning of the sham stimulation, while preventing any effective modulation of cortical excitability by sham tDCS, and thus allowing for a successful blinding of participants for the real vs. sham stimulation condition [56]. The study was a single-blind experiment: participants were not aware of the type of stimulation they received, whereas the experimenter was fully informed (see [1] for a similar procedure). Concurrently with the beginning of the stimulation, a cartoon movie was projected on the computer screen. This was done reduce inter-subjects’ variability by exposing participants to the same visual experience during the stimulation period (ibidem). After 18 min since the beginning of the stimulation, the cartoon movie was stopped, and participants were told that in 2 min they would have to perform the rating task for a new set of musical excerpts. The rating task was administered within 1 min from the end of the tDCS stimulation. All participants completed the task within 10 min from the end of the tDCS stimulation.

Before starting, instructions were repeated to participants. Excerpts were presented in a random fixed order. The whole experimental session lasted approximately 45 min. The software E-prime 2.0 (Psychology Software Tools, Inc., Pittsburgh, PA, USA) was used for stimuli presentation and data recording.

## 3. Results

### 3.1. Experiment #1

#### 3.1.1. Esthetic Evaluation

##### Rating Scores

Figure 2 shows the mean percentage rating scores for the esthetic evaluation by *Group* (musicians and non-musicians) and *Judgement* (“beautiful”, “medium”, and “ugly”). A repeated-measures analysis of variance (ANOVA), with *Judgment* (“beautiful”, “medium”, and “ugly”) as a within-subjects main factor and *Group* (musicians and non-musicians) as a between-subjects main factor, was carried out on the mean percentage rating scores for the esthetic evaluation (see Table 1). The analysis revealed: (i) a significant main effect of *Judgment*; (ii) no significant main effect of *Group* (musicians vs. non-musicians); (iii) a trend towards significance for the interaction group × judgment, indicating that the effect of group varied at different levels of judgment. As for the main effect, the differences among the three levels of judgment were then analyzed by multiple comparisons applying the Bonferroni correction. Significant differences between “beautiful” vs. “ugly” (*p* < 0.001), “beautiful” vs. “medium” (*p* < 0.001), and “medium” vs. “ugly” (*p* < 0.001) were found. Regarding the interaction comparisons, musicians reported a lower score for “beautiful” (M ± SD = 62.96 ± 11.54% for musicians; M = 65.03 ± 10.33% for non-musicians), and a higher score for “medium” (M ± SD = 57.17 ± 12.92% for musicians; M ± SD = 52.46 ± 8.97% for non-musicians) and for “ugly” (M ± SD = 46.14 ± 14.9% for musicians; M ± SD = 43.95 ± 9.7% for non-musicians) judgments.

Bayesian analysis indicated that data provided very strong support for the inclusion of *Judgment* as a predictor, BF_incl_ = 8.75 × 10^16^, but only very weak evidence for excluding the main effect of *Group* (BF_incl_ = 0.47) or including the interaction effect (BF_incl_ = 1.05). Post hoc comparisons for the *Judgment* effect provided evidence indicating different rating scores between the “beautiful” and “ugly” (BF_10_ = 2.99 × 10^11^), “beautiful” and “medium” (BF_10_ = 87,652.5), and “ugly” and “medium” (BF_10_ = 1.045 × 10^6^) judgment levels.

##### Response Latencies

Figure 3 shows the mean response latencies (RT) for the esthetic evaluation, by *Group* (musicians, non-musicians) and *Judgment* (“ugly”, “beautiful”, and “medium”). A repeated-measures ANOVA, with *Judgment* as a within-subjects main factor, and *Group* as a between-subjects main factor, was performed (see Table 1). The analysis revealed: (i) a significant effect of the main judgment factor; (ii) a significant main effect of *Group*; (iii) but no significant interaction effect *Judgment × Group*. As for the *Judgment* factor, post hoc analyses (Bonferroni correction applied) indicated only a significant difference between “beautiful” and “ugly”, *p* = 0.04 (“beautiful” vs. “medium”, *p* = 0.22; “ugly” vs. “medium”, *p* = 1), with “beautiful” excerpts obtaining the fastest times (M ± SD = 1465 ± 67 msec), followed by “medium” (M ± SD = 1559 ± 86 msec), and “ugly” (M ± SD = 1617 ± 99 msec) tracks. Regarding the *Group* factor, musicians proved to be slower (M ± SD = 1704 ± 607 msec), as compared to non-musicians (M ± SD = 1389 ± 338 msec), in making an esthetic evaluation of musical excerpts.

Bayesian analysis indicated that data provided anecdotal evidence for the inclusion of *Judgment*, BF_incl_ = 1.35, and *Group*, BF_incl_ = 1.69 as predictors, and moderate evidence for excluding the interaction effect (BF_incl_ = 0.12). Post hoc comparisons for the *Judgment* effect provided evidence indicating that reaction times were faster in the “beautiful” than in the “ugly” condition (BF_10_ = 3.36), but not when the “beautiful” and “medium” (BF_10_ = 0.80), and the “ugly” and “medium” Judgment (BF_10_ = 0.24) were compared.

In sum, the results demonstrated significant differences for all participants for all three categories: “beautiful”, “ugly”, and “medium”. The “beautiful” excerpts obtained the highest scores, the “ugly” ones the lowest scores, and the “medium” ones were placed at an intermediate level. As for musicians and non-musicians, the present data did not show significant differences in the judgements of the two groups about the excerpts. However, musicians were significantly slower than non-musicians.

#### 3.1.2. Emotional Evaluation

##### Rating Scores

Figure 4 shows the mean percentage rating scores for the emotional evaluation, by *Group* (musicians and non-musicians) and *Judgment* (beautiful, medium, and ugly). In line with the results obtained in the Esthetic evaluation, the “beautiful” category obtained the highest rating, followed by the “medium” and lastly by the “ugly”.

A repeated-measures ANOVA with *Judgment* (“beautiful”, “medium”, and “ugly”) as the within-subjects main factor and *Group* (musicians and non-musicians) as the between-subjects main factor was carried out (see Table 1). The analysis revealed: (i) a significant effect of the main factor of *Judgment*; (ii) no significant main effect of *Group*; and (iii) no significant interaction effect of *Judgment* by *Group*. As for the main effect of *Judgment*, Bonferroni-corrected multiple comparisons demonstrated significant differences between the three categories (all *Ps* < 0.001). 

Bayesian analysis indicated that data provided very strong support for the inclusion of *Judgment* as a predictor, BF_incl_ = 2.67 × 10^28^, but only very weak evidence for excluding the main effect of *Group* (BF_incl_ = 0.46), and moderate evidence for excluding the interaction effect *Judgment* by *Group* (BF_incl_ = 0.18). Post hoc comparisons for the *Judgment* effect provided evidence indicating that the emotional mean rating score was higher for the “beautiful” compared to both the “ugly” (BF_10_ = 3.05 × 10^18^) and the “medium” judgment (BF_10_ = 2.96 × 10^11^), and for the “medium” compared to the “ugly” judgment (BF_10_ = 241,556.51 × 10^6^).

##### Response Latencies

Figure 5 shows the mean response latencies (RT) for the emotional evaluation as a function of *Group* (musicians and non-musicians) and *Judgment* (“beautiful”, “medium”, and “ugly”). A repeated-measures ANOVA with *Judgment* as a within-subjects main factor and *Group* as a between-subjects main factor revealed a significant main effect of *Judgment*, and a main effect of *Group*. The *Judgment* by *Group* interaction was not significant. As for the main effect of *Judgment*, multiple comparisons, with the Bonferroni correction applied, showed significant differences between the “beautiful” vs. “ugly” (*p* < 0.001) and between the “medium” vs. “ugly” (*p* = 0.04) judgements; the difference between the “beautiful” vs. “medium” judgements was not significant (*p* = 0.15). “Beautiful” excerpts obtained the fastest reaction times (M ± SD = 1584 ± 94 msec), followed by “medium” (M ± SD = 1684 ± 105 msec) and “ugly” tracks (M ± SD = 1826 ± 125 msec). As for the differences between the groups, musicians were slower in judging the emotional valence of musical excerpts (M ± SD = 1989 ± 863.9 msec), compared to non-musicians (M ± SD = 1407 ± 353 msec).

Bayesian analysis indicated that data provided very strong support for the inclusion of *Judgment* as a predictor, BF_incl_ = 242.25, and moderate evidence for including the main effect of *Group*, BF_incl_ = 4.63. The BF for the *Judgment* by *Group* interaction was 0.19, indicating moderate evidence for excluding the interaction effect as the predictor. Post hoc comparisons for the *Judgment* effect provided evidence indicating faster reaction times for the “beautiful” compared to the “ugly” judgment (BF_10_ = 306.3), and for the “medium” compared to the “ugly” judgment (BF_10_ = 3.09). Results indicated evidence that reaction times of “beautiful” and “medium” judgment were the same (BF_10_ = 1.12).

#### 3.1.3. Brief Discussion

Overall, Experiment #1 had a double aim: firstly, to assess whether there was any difference in the esthetic and emotional evaluation of musical excerpts between musicians and non-musicians; secondly, to set up a task, providing a useful tool, to assess “liking” and emotions in music, both for music experts and naïve individuals. As for the emotional and esthetic experience, the results demonstrated significant differences for all participants in both tasks, and for all three categories: “beautiful”, “ugly”, and “medium”. For both the esthetic and the emotional judgments, the “beautiful” excerpts obtained the highest scores, the “ugly” ones the lowest scores, and the “medium” ones were placed at an intermediate level.

As for the differences between musicians and non-musicians, the present data do not show significant differences in the judgements of the two groups about the excerpts, both emotionally and esthetically: the recorded judgements are then independent of musical expertise. On the other hand, musicians are significantly slower than non-musicians.

Overall, the results highlight a correlation between esthetic and emotional evaluations in music after a short listen (i.e., seven seconds), demonstrating that higher levels of beauty correspond to more positive emotions. Furthermore, this correlation is independent of a previous musical training, which, in contrast, seems to influence only the time needed to express the judgment, suggesting a more analytic process for higher levels of expertise.

### 3.2. Experiment 2

#### 3.2.1. Comparison between Esthetic Rating Scores of Experiments #1 and #2

To evaluate the robustness of the results of Experiment #1 about esthetic judgments, the esthetic scores obtained by non-musicians in Experiment #1 were compared with those obtained in Experiment #2 by the participants who scored stimuli of Set #1 in the pre-stimulation stage. Three independent sample *t*-tests were carried out between the two groups, for each category of stimuli: “beautiful”, “medium”, and “ugly”. No significant differences were found (“beautiful” *p* = 0.44; “ugly” *p* = 0.13; “medium” *p* = 0.41).

#### 3.2.2. tDCS Side Effects

After each stimulation, a questionnaire was administered to all participants, to detect the presence of any side effect or discomfort sensations. Transient discomfort sensations as itch or light burning on the scalp were sometimes (N = 5 for the “sham” condition; N = 8 for the “real” condition) reported during the initial phases of either stimulation. 

#### 3.2.3. Rating Scores

As preliminary analyses demonstrated no significant differences between the evaluations of set #1 and set #2, in all subsequent analyses, the participants were considered as a single group (N = 22). 

The position of the mouse cursor along the bar was automatically converted by the software to percentage rating scores, where a 0% score corresponded to the mouse cursor positioned at the lower end (red) of the rating bar, and a 100% score to the mouse cursor positioned at the upper end (green) of the rating bar. Analyses were performed on mean rating scores.

Figure 6 shows the percentage of esthetic judgment as a function of session (pre- and post-tDCS), tDCS condition (real and sham) and judgment (“beautiful”, “medium”, and “ugly”). A repeated-measures ANOVA, with *tDCS Condition* (sham and real), *Session* (pre- and post-tDCS) and *Judgment* (“beautiful”, “medium”, and “ugly”) as within-subject main factors, demonstrated no significant main effect of the *tDCS Condition* (see Table 2). On the contrary, the main effects of *Session* were significant, exhibiting a decrease of mean ratings from pre-tDCS (M ± SD = 51.98 ± 10.55%) to post-tDCS (M ± SD = 49.82 ± 11.65%). Furthermore, the main effect of *Judgement* and the *Session* by the *Judgment* interaction were also significant. Nevertheless, the *tDCS Condition* by *Session*, the *tDCS Condition* by *Judgment,* and the *tDCS Condition* by *Session* * *Judgment* interactions were not significant. As for the main effect of *Judgment*, the Bonferroni-corrected multiple comparisons showed significant differences between the three categories (all *Ps* < 0.001), with a higher mean score for the “beautiful” category (M ± SD = 64.93 ± 12.57%), as compared to the “medium” (M ± SD = 54.52 ± 12.96%) and the “ugly” (M ± SD = 33.25 ± 13.79%) one, which obtained the lowest rating. As for the Session × Judgment interaction, mean scores indicate that the “beautiful” (M ± SD = 65.36 ± 11.97% pre-tDCS, M ± SD = 64.5 ± 13.26% post-tDCS) and the “medium” (M ± SD = 55.39 ± 12.54% pre-tDCS, M ± SD = 53.66 ± 13.45% post-tDCS) categories, but not the “ugly” (M ± SD = 35.2 ± 13.07% pre-tDCS, M ± SD = 31.3 ± 14.35% post-TDCS) category, obtained higher scores during the pre-tDCS condition as compared to the post one.

Bayesian analysis indicated that the data provided very strong support for the inclusion of *Judgment* as a predictor, BF_incl_ = 4.91 × 10^67^. The post hoc tests indicated overwhelming evidence that the “liking” rating score differed between “beautiful” and “medium” judgment (BF_10_ = 2.98 × 10^11^), between “beautiful” and “ugly” judgment (BF_10_ = 3.10 × 10^32^), and between “medium” and “ugly” judgment (BF_10_ = 6.61 × 10^27^). The data also indicated weak evidence for the inclusion of *Session* as a predictor, BF_incl_ = 1.12. Finally, the data indicated moderate evidence for excluding the *tDCS Condition* as predictor, BF_incl_ = 0.21, as well as all the interactions effects (Session × tDCS Condition: BF _incl_ = 0.18; Session × Judgment: BF _incl_ = 0.13; TMS condition × Judgment: BF _incl_ = 0.12; Session × tDCS condition × Judgment: BF _incl_ = 0.14).

#### 3.2.4. Response Latencies

In line with the procedure followed for the rating scores, participants were considered as a single group including 22 subjects.

Figure 7 shows the mean reaction times as a function of session (pre- and post-tDCS), tDCS condition (real and sham) and judgment (“beautiful”, “medium”, and “ugly”). A repeated-measures ANOVA, with *tDCS Condition* (sham and real), *Session* (pre- and post-tDCS) and *Judgment* (beautiful, medium, and ugly) as within-subject main factors, demonstrated no significant main effect of *Condition* (see Table 2). The main effect of *Session* was significant, with increased response latencies during the pre-tDCS task (M ± SD = 1427 ± 541 msec) than during the post-tDCS task, (M ± SD = 1322 ± 518 msec). Moreover, the main effect of *Judgment* was significant. All interaction effects were not significant: *Condition* × *Session*, *Condition* × *Judgment*, *Session* × *Judgment* and *Condition* × *Session* × *Judgment*. The differences among the three levels of *Judgment* were analyzed by Bonferroni-corrected multiple comparisons; significant differences were found between the “beautiful” vs. “ugly” (*p* = 0.01) and the “medium” vs. “ugly” (*p* = 0.02), with the “beautiful” vs. “medium” comparison being not significant (*p* = 0.89). Particularly, participants were faster for “beautiful” excerpts (M ± SD = 1310 ± 104 msec), as compared to the “medium” tracks (M ± SD = 1360 ± 109 msec) and to the “ugly” ones (M ± SD = 1453 ± 106 msec), which required the longest times.

Bayesian analysis indicated moderate evidence for including both *Session*, BF_incl_ = 4.80 and *Judgment*, BF_incl_ = 3.36, as predictors. Post hoc tests provided evidence that latencies differed between the “beautiful” and the “ugly” judgments (BF_10_ = 153.45) and between the “medium” and the “ugly” ones BF_10_ = 8.37). Conversely, the latencies of “beautiful” and “medium” judgment did not differ from each other (BF_10_ = 0.24). Analyses also indicated moderate evidence for excluding the *tDCS Condition* as predictor, BF_incl_ = 0.29, as well as the interaction effects *Session* × *tDCS Condition*: BF _incl_ = 0.35, and *Session* × *tDCS Condition* × *Judgment*: BF _incl_ = 0.17. Finally, data indicated strong evidence for excluding as predictors both the *Session* × *Judgment*, BF _incl_ = 0.07, and the *tDCS Condition* × *Judgment*, BF _incl_ = 0.07, interactions. 

#### 3.2.5. Brief Discussion

Experiment #2 investigated the role of the left DLPFC in the esthetical judgment of musical pieces, using the tDCS. Contrary to the expectations and previous findings in the field of visual art [1], the results do not show any significant effect of tDCS, suggesting that the stimulation of the left DLPFC is not enough to modulate musical taste after a short listen. 

Concerning the ratings, the results show a significant difference between pre- and post-tDCS scores, independent of the type of stimulation (real or sham), with rates being lower after tDCS. This result may be accounted for in terms of a generalized effect due to the participants’ fatigue and the repetition of the task.

Finally, the pattern of response latencies, with faster responses in the second (post-tDCS) part, with respect to the first (pre-tDCS) part of the study, would likely reflect a stabilization of the response criterion used and a task familiarization effect, resulting in faster responses.

## 4. General Discussion

Experiment #1 highlighted how esthetic judgment and emotional valence follow the same trend in music, both reporting higher scores for excerpts previously classified as “beautiful” and lower scores for the “ugly” tracks.

The classification of the stimuli made before the experiments, together with the results of the present study, support the hypothesis that esthetics judgment and emotional valence generally correlate in music. In fact, previous findings show that the music preferences of both musicians and non-musicians vary only in a few music genres, such as classical and jazz music, which are preferred by musicians [57]. However, considering music taste on a whole level, not referring to any specific genre, both musicians and non-musicians exhibit a preference for consonant intervals, clear timbre, tonal structure, and regular rhythms [58]. This trend seems to be culturally influenced, as Amazonian natives show indifference toward dissonance sounds [59]. Nevertheless, in western populations, these preferences can be verified since childhood: 6-month-old infants, exposed to occidental musical culture, prefer consonant sounds [60]. Although infants do not yet have a musical-system-specific knowledge of scale structure, which is involved in adults’ emotional reactions to music, they resemble adults in their evaluative reactions to consonance and dissonance. The presence of a significant interaction effect between esthetical and emotional evaluations indicates that participants perceive more positive emotions for those excerpts judged as “beautiful”, and more negative emotions for those excerpts judged as “ugly”. However, there is evidence that liking and emotional valence can be evaluated with divergent outcomes, by both musicians and non-musicians, who may judge an excerpt as “beautiful”, but still not be emotionally moved by it [33,61]. The tracks used in those studies [33,61], however, were much longer than those used for the current one, then possibly providing more time for higher-order cognitive processes of elaboration to occur; this, in turn, would have resulted in a more complex relationship between judgements of “liking” and emotion al valence. Keeping a seven-sec duration for each musical excerpt permits the exploration of the “first” feeling and judgement of participants. In this respect, the results of Experiment #1 suggest that the same task may be used to assess both components, as there is an average correspondence between the two types of judgement; the task is also appropriate to detect dissociations between the two judgements.

Furthermore, esthetic judgment and emotional valence in music seem to be independent of the level of musical training, adding more evidence to previous findings showing a homogeneity of the answers of musicians and non-musicians about esthetic judgment of chord sequences, with no differences between expert and naïve participants [62]. As for latencies, musicians are significantly slower than non-musicians. This result, which at first sight may appear counterintuitive, could be explained by the hypothesis that, while listening to the excerpts, musicians adopt a more analytic modality of judgment, due to their musical training, compared to the more global one used for an esthetic judgment by non-musicians [7,10,63]. In line with this view, a more analytic analysis may require a major cognitive effort, resulting in an increase of response latencies. An electrophysiological counterpart of this hypothesis comes from the report in musicians of a stronger contingent negative variation (CNV), compared to non-expert participants, when they are required to provide an esthetic, rather than a correctness, judgment about a piece of music [64]. CNV is a slow negative potential with a shallow slope, which correlates with task difficulty, as well as with the amount of effort allocated to task planning and execution [30]; accordingly, esthetic judgment may require a greater effort for musicians than to non-musicians. 

Experiment #2 does not demonstrate any significant effect of tDCS stimulation. Cattaneo and colleagues [1] performed a study similar to the present one, focusing instead on visual art. An anodal tDCS was placed over the left DLPFC, with the result of enhancing the esthetical judgment on visual stimuli, which obtained higher “liking” scores by participants after tDCS. Interestingly, the variation of ratings concerned only the figurative paintings, but not abstract art, suggesting that the neural mechanisms underlying the appreciation of figurative and abstract images may be different (at least in individuals with no strong background in fine arts, who tend to spontaneously prefer figurative images). Zatorre and Salimpoor [65] consider music as “the most abstract art”, as its esthetic appeal has little to do with recounting events or depicting people, places, or objects, which are the province of the verbal and visual arts. A sequence of pitches—such as might have been produced by an ancient flute—concatenated in a certain way, cannot specifically denote anything, but can certainly result in emotions [66]. Hence, observed from this perspective, the present data would be in line with those by Cattaneo and colleagues [1], who found no effects of tDCS with abstract paintings. Abstract art, unlike figurative art, may require knowledge not available to individuals naïve to visual art, to be fully appreciated [1]. This could negatively affect the possibility of modulating its perception. Furthermore, paintings were shown entirely, while the stimuli adopted in the present research were just scraps of musical pieces. The possibility may be also entertained that stimulating only the left DLPFC may be not enough to significantly modulate the esthetic experience of music, as a more extensive neural network is involved in this process. Blood and Zatorre [67] recorded cerebral activity through positron emission tomography (PET) in non-expert participants while they listened to music, finding a pattern of activation in a wide network, including the left ventral striatum, the left dorsal mid-brain, the amygdala (bilaterally), and the right orbito-frontal and ventro-medial prefrontal cortices.

Furthermore, music preferences may vary widely among individuals, depending on several modulating factors, including: (i) person-specific features or the “internal context”, including expertise, internal state, mood, personality, and attitude; (ii) factors related to the listening situation, also described as “external context”, including the physical and social environment, such as being at home or elsewhere, as in a concert hall; being alone or with others) [68,69]. Therefore, as highlighted by Cattaneo and colleagues [4] for visual art, and also in music, the effects of several contextual aspects, including the individual educational background and the individual’s expertise in the processing of the listened stimuli on their appreciation, remain to be investigated, as well as the effects of these variables on the recruitment of different brain networks during esthetic appreciation.

Ratings, independent of the type of stimulation (real or sham), are lower after tDCS. This result may be accounted for in terms of a generalized effect due to the participants’ fatigue and the repetition of the task. Fatigue may be an interfering variable in the evaluation of pleasantness, as observed in other areas, such as food pleasantness ratings [70].

Finally, the pattern of response latencies, with faster responses in the second (post-tDCS) part, with respect to the first (pre-tDCS) part of the study, may reflect a stabilization of the response criterion used and a task familiarization effect, resulting in faster responses, as observed in previous studies [1].

On the other hand, overall, the results obtained in Experiments #1 and #2 have to be considered along with a few limitations aspects: first, the present samples included only relatively young participants; second, music excerpts were not chosen by the participants themselves. Finally, future studies should also specifically explore the influence of different music genres on the esthetic judgment of music.

## 5. Conclusions

In summary, the present findings show that esthetic and emotional judgments agree when people (in the present study, all below 35 years of age) are asked to rate their first impression about music tracks, independent of their musical expertise. Furthermore, musicians tend to be slower than laypersons, possibly indicating that they allocate comparatively more resources and effort when required to express an esthetic judgment about music. The present results, unlike previous findings in visual art [1], demonstrate no significant effects of a single tDCS session on esthetic and emotional judgments. With the caution that conclusions are based on negative findings, the present results suggest that stimulating the left DLPFC through tDCS is not enough to bring about detectable changes of the partcipants’ esthetic appreciation of music.

## Figures and Tables

**Figure 1 brainsci-12-00467-f001:**
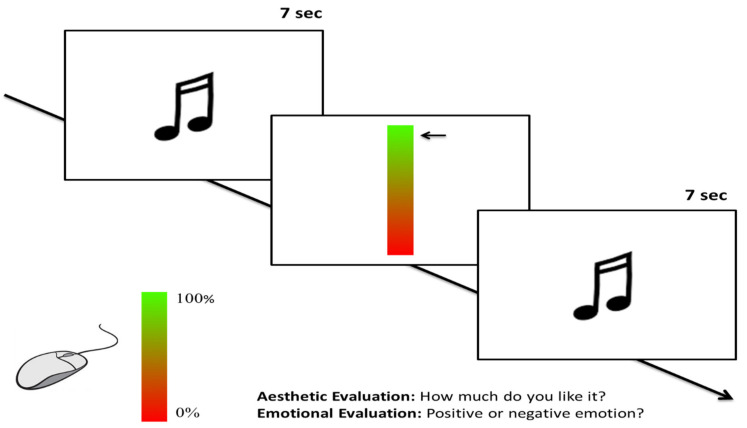
Example of an experimental trial. In each trial, an excerpt was presented while the screen was totally white. The participants’ task was to indicate, by moving the mouse cursor along a rating bar, how much they liked the track (esthetic evaluation) or how much they felt a positive or negative emotion (emotional evaluation). The lower red end of the rating bar corresponded to 0% score (i.e., “I do not like it at all”/“entirely negative emotion”). The upper green end of the rating bar corresponded to 100% score (“I entirely like it”/“entirely positive emotion”). The bar remained visible until participants expressed their judgment. Then, after 300 ms, a new track was played.

**Figure 2 brainsci-12-00467-f002:**
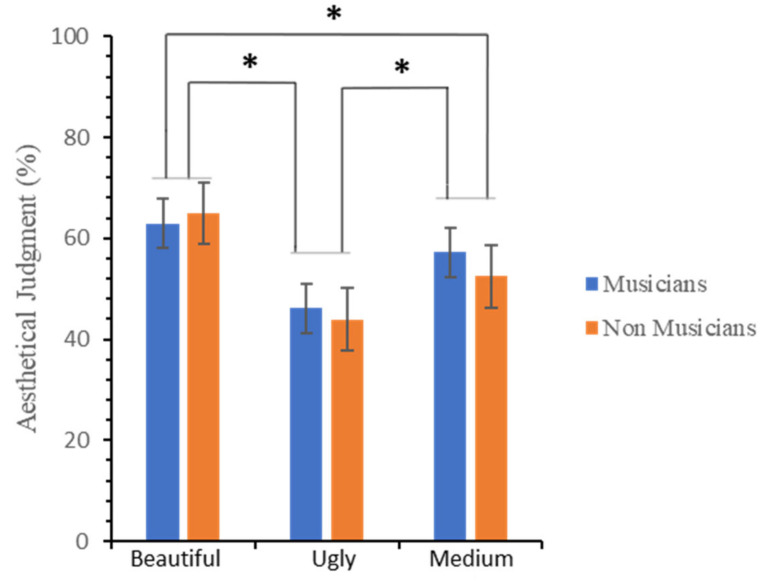
Participants’ mean rating percentage scores for the esthetic evaluation. Error bars represent ± 1 SEM. Asterisks represent significant differences.

**Figure 3 brainsci-12-00467-f003:**
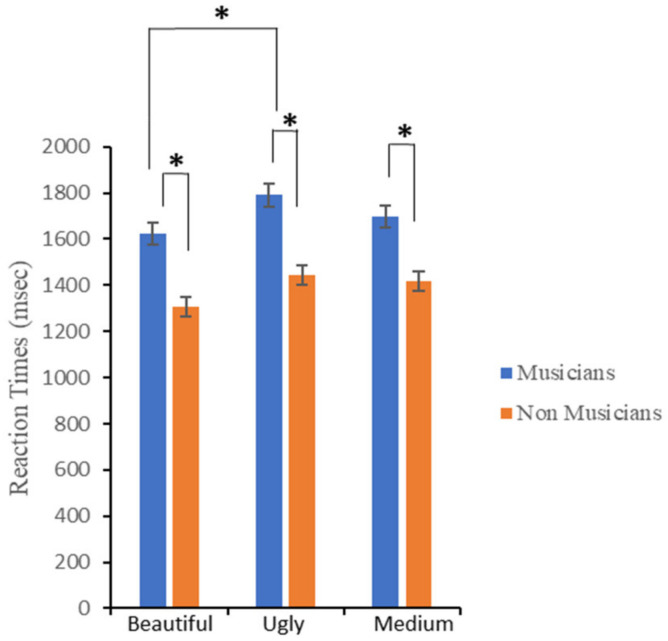
Participants’ mean reaction times (ms) for the esthetic evaluation. Error bars represent ± 1 SEM. Asterisks represent significant differences.

**Figure 4 brainsci-12-00467-f004:**
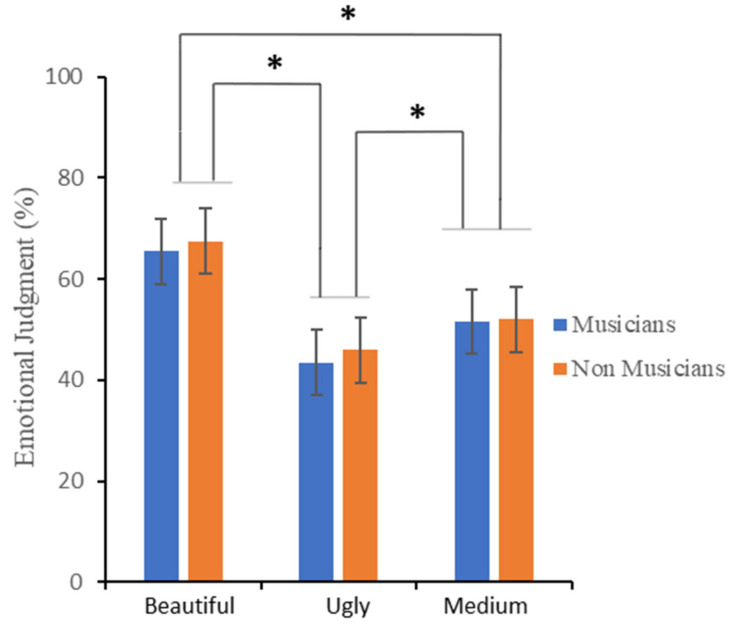
Participants’ mean rating percentage scores for the emotional evaluation. Error bars represent ± 1 SEM. Asterisks represent significant differences.

**Figure 5 brainsci-12-00467-f005:**
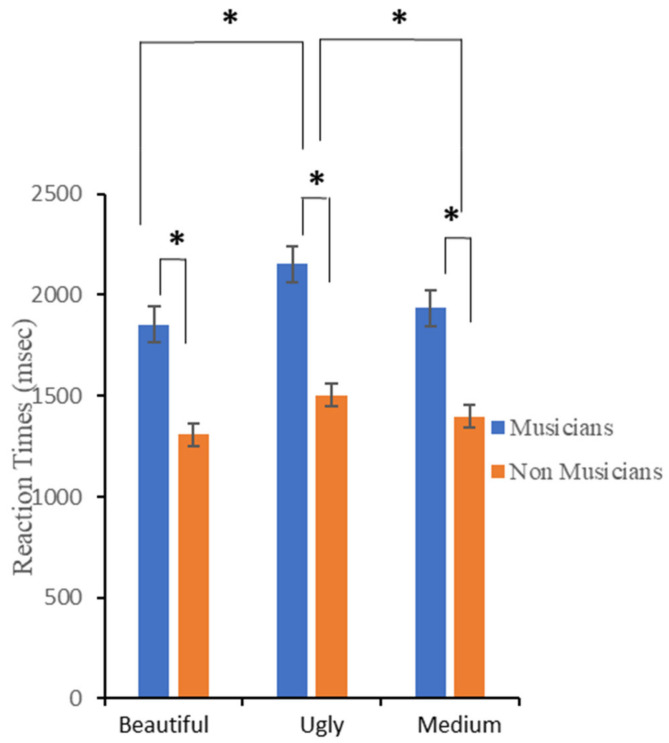
Participants’ mean reaction times (ms) for the emotional evaluation. Error bars represent ± 1 SEM. Asterisks represent significant differences.

**Figure 6 brainsci-12-00467-f006:**
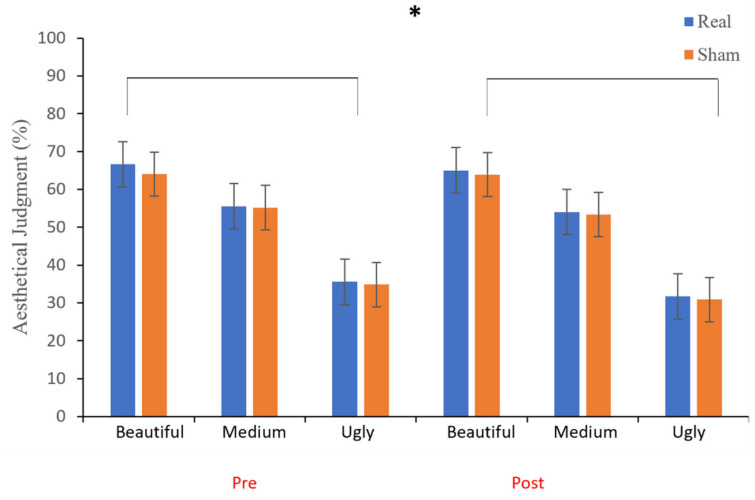
Participants’ mean rating percentage scores pre- and post- tDCS, both for sham and real conditions. Error bars represent ± 1 SEM. * represents significant differences.

**Figure 7 brainsci-12-00467-f007:**
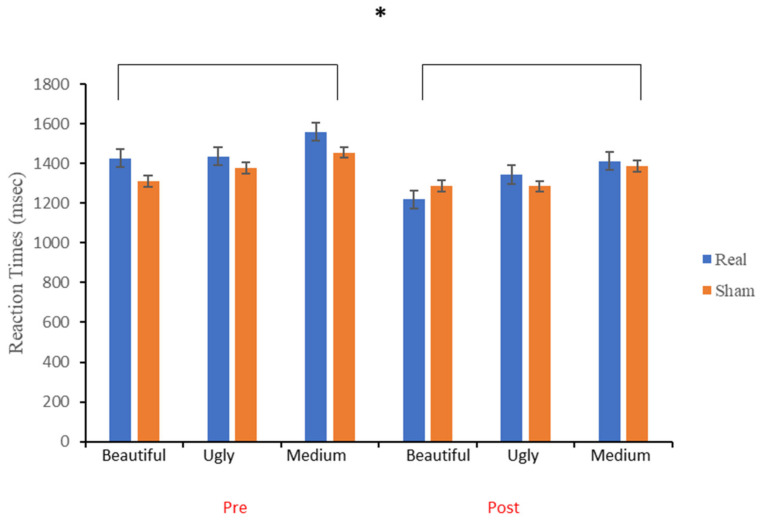
Participants’ mean reaction times (msec) pre- and post-tDCS, for the sham and real conditions. Error bars (±1 SEM). * (significant differences).

**Table 1 brainsci-12-00467-t001:** Main statistic results of Experiment #1. Significant *p* values are reported in bold.

Effects (Liking—Rating Scores)	Statistics Results	*p* Values
Judgment	F (2, 76) = 87.16	***p* < 0.01**, ηp^2^ = 0.69
Group	F (1, 38) = 0.24	*p* = 0.63, ηp^2^ = 0.01
Group × Judgment	F (2, 76) = 2.85	***p* = 0.06**, ηp^2^ = 0.07
**Effects (Liking—Response Latencies)**		
Judgment	F (2, 76) = 3.35	***p* = 0.04**, ηp^2^ = 0.08
Group	F (1, 38) = 4.11	***p* = 0.05**, ηp^2^ = 0.1
Group × Judgment	F (2, 76) = 0.16	*p* = 0.85, ηp^2^ = 0.004
**Effects (Emotion—Rating Scores)**		
Judgment	F (2, 76) = 184.1	***p* < 0.01**, ηp^2^ = 0.83
Group	F (1, 38) = 0.57	*p* = 0.46, ηp^2^ = 0.02
Group × Judgment	F (2, 76) = 0.45	*p* = 0.65, ηp^2^ = 0.011
**Effects (Emotion—Response Latencies)**		
Judgment	F (2, 76) = 10.5	***p* < 0.01**, ηp^2^ = 0.22
Group	F (1, 38) = 7.78	***p* = 0.01**, ηp^2^ = 0.17
Group × Judgment	F (2, 76) = 0.62	*p* = 0.54, ηp^2^ = 0.02

**Table 2 brainsci-12-00467-t002:** Main statistic results of Experiment #2. Significant *p* values are reported in bold.

Effects (Rating Scores)	Statistics Results	*p* Values
Condition	F (1, 21) = 1.02	*p* = 0.32, ηp^2^ = 0.05
Session	F (1, 21) = 8.47	***p* < 0.01**, ηp^2^ = 0.29
Judgment	F (1, 21) = 92.88	***p* < 0.001** ηp^2^ = 0.82
Session × Judgment	F (1, 21) = 4.51	***p* = 0.017** ηp^2^ = 0.31
Condition × Session	F (1, 21) = 0.08	*p* = 0.79, ηp^2^ < 0.01)
Condition × Judgment	F (1, 21) = 0.72	*p* = 0.5 ηp^2^ = 0.07
Session × Condition × Judgment	F (1, 21) = 0.57	*p* = 0.57 ηp^2^ = 0.05
**Effects (Response Latencies)**		
Condition	F (1, 21) = 0.55	*p* = 0.47, ηp^2^ = 0.03
Session	F (1, 21) = 4.49	***p* = 0.046**, ηp^2^ = 0.18
Judgment	F (1, 21) = 6.35	***p* < 0.01**, ηp^2^ = 0.23
Session × Judgment	F (1, 21) = 0.07	*p* = 0.93, ηp^2^ = 0.003
Condition × Session	F (1, 21) = 0.89	*p* = 0.36, ηp^2^ = 0.04
Condition × Judgment	F (1, 21) = 0.18	*p* = 0.83, ηp^2^ = 0.01
Session × Condition × Judgment	F (1, 21) = 0.91	*p* = 0.41, ηp^2^ = 0.04

## Data Availability

Data can be requested from the corresponding authors.

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
