# Peer review of "Exploring the Effects of Brain Stimulation on Musical Taste: tDCS on the Left Dorso-Lateral Prefrontal Cortex—A Null Result"

_brainsci, 2022, doi:10.3390/brainsci12040467_

Round 1
Reviewer 1 Report
The paper "Exploring the effects of brain..." is complete and was interesting to read. It is very commendable to report a null result. This reviewer has two major concerns:
(i) Comparing Fig. 6 to Fig. 2 the ugly gets the lowerst percentage in Fig. 2, but the medium gets the lowest percentage in Fig. 6. Is that correct?
(ii) In the experiment 2 it is not clear that in the second session the subjects are naive towards the stimuli. Since you have only two sets of stimuli used already in the first session subjects know the stimuli in the second session 3.5 days after the first session. The description is confusing here, combine the first paragraph in 2.2.3 with the 3rd paragraph in this section. Maybe that clarifies mattters or add a picture.
Minor concerns:
Reading the statistics results for each figure is extremely tedious. Can this be combined for all (?) figures into a table and only the important results described in the text?
More details could be given for the pilot study establishing the categories.
The ad hoc set #2 is created from the same pieces of music as #1. Is that correct? Probably state more clearly.
Author Response
Thank you for your precious suggestions and constructive comments.
(i) Comparing Fig. 6 to Fig. 2 the ugly gets the lowest percentage in Fig. 2, but the medium gets the lowest percentage in Fig. 6. Is that correct?
We would like to thank the reviewer for noticing this point. Indeed, there was a mistake in Fig. 6, which is now corrected in the new version of the manuscript (line 533).
(ii) In the experiment 2 it is not clear that in the second session the subjects are naive towards the stimuli. Since you have only two sets of stimuli used already in the first session subjects know the stimuli in the second session 3.5 days after the first session. The description is confusing here, combine the first paragraph in 2.2.3 with the 3rd paragraph in this section. Maybe that clarifies matters or add a picture.
Lines 311-313 were removed in order not to create misunderstandings or doubts about the order of presentation of the stimuli. Furthermore, specifications were added (lines 255-256).
(iii) Reading the statistics results for each figure is extremely tedious. Can this be combined for all (?) figures into a table and only the important results described in the text?
Thank you for your precious suggestion. We added Table 1 (line 349) and Table 2 (line 511) in order to sum up most of the statistics results.
(iv) More details could be given for the pilot study establishing the categories.
More details were added in lines 157-158.
(v) The ad hoc set #2 is created from the same pieces of music as #1. Is that correct? Probably state more clearly.
We thank the reviewer for raising up this point. We agree that this can create confusion. A clearer sentence has been added in the new version of the manuscript (lines 240).
Reviewer 2 Report
Thanks for giving me the opportunity to review this paper (ID = 1601138). In two experiments, the authors study whether expertise has any effect on aesthetic preferences for music. In Experiment 1, they address this question by comparing liking and emotional ratings in two groups: musicians vs non- musicians. In Experiment 2, they use tDCS on a group of non-musicians to examine whether stimulation on the left DLPFC would result in increased aesthetic appreciation. Results from the two experiments show that expertise (musicians vs non-musicians) does not have any impact on aesthetics ratings of music (Experiment 1), nor does tDCS applied on an area of the brain involved in aesthetic appreciation (Experiment 2). While I value the work that the authors put into this manuscript, I have major concerns that prevent me from suggesting this work for publication. I list them below:
- The rationale of the study and clarity: It was not clear to me what is the rationale behind the two experiments reported in this paper. Moreover, the introduction does not really help in clarifying those. In particular:
- Why did the authors expect in the first place any differences in the aesthetic responses of music by expert musicians and non-musicians? The fact that there exist differences in the auditory sensorimotor and subcortical regions in the two groups, does not necessarily mean they should have different aesthetic responses to music. The authors should clarify this in the first paragraph of the introduction when explaining the motivation of Experiment 1.
- Similarly, what is the connection between Experiment 1 and Experiment 2? Experiment 1 examines the effect of music expertise on aesthetic responses to music, whereas Experiment 2 studies the role of a particular brain region involved in aesthetic appreciation. This should also be clarified in the paper.
- The introduction would benefit from stating clearly the hypotheses that guided this work. This would also help to improve the clarity in the results section.
- In terms of clarity and fluidity, it would help to present first the methods and results of Experiment 1, and then do the same for Experiment 2 (potentially adding a short discussion after each experiment). Currently, presenting the methods of the two experiments in succession and then the same for the results is a bit confusing.
- Some information seems to be repeated, for example in the procedure of Experiment 2 (see line 49 and line 291).
- Some parts of the manuscript would benefit from English proofreading.
- Contribution and methodological limitations: Overall, it’s not clear what is the contribution of this work. Is the contribution that the null results suggest different brain networks for aesthetic appreciation in music and art. If so, this is a very strong claim based on the experiments and results reported in the paper. On the other hand, the contribution may be that the study shows that there are no differences in aesthetic appreciation depending on expertise. In this case, I’m not convinced that this is a very important contribution by itself, but perhaps I could be convinced otherwise (e.g., why would we expect differences based on expertise? Is this a common finding in other domains?). Importantly, the observed results could be explained by several limitations of the design and methods used in the experiments. Specifically:
- Measuring musicality: it seems that the authors only used “previous music training” to classify participants as non-musicians. However, musicality is a complex construct that depends on several other dimensions other than formal training. For example, a DJ or music blogger may know a lot about music without having any previous formal training. The golden standard method in music cognition to measure musical expertise, the GOLD-MSI, identifies 5 musicality dimensions: musical training, active engagement to music, perceptual abilities, singing abilities, and music emotionality. In a study where the differences between musicians and non-musicians are so important, it would help a lot to understand better how different the two groups really were in terms of their musicality. I strongly recommend to the authors using this tool in the future to measure music expertise (see reference below, Müllensiefen et al., 2014):
- Similarly, there are validated tools to study emotional reactions to music, which is a complex and huge area of research in music psychology and music cognition (not measuring emotions evoked by music properly is, therefore, an important problem). See for example Zentner et al. 2008 (full reference below).
- Musicians were slower than non-musicians: the authors argue that this result may be explained by the idea that musicians’ judgements were more analytical and required higher cognitive effort. However, the results could alternatively be explained by the fact that musicians enjoyed the music listening task more than non-musicians and therefore were on average slower.
- Rating task and stimuli: The null effects found between the two groups could be explained by the experimental task and stimuli used. The authors did a good job in sampling the music stimuli and including a wide variety of sounds representing music. However, it is likely that if differences in aesthetic responses to music exist in the two groups, these are concentrated in particular music genres or styles (e.g., classical music for formally trained musicians or jazz music for jazz musicians). Thus, controlling for and specifically studying different music genres is an important step (perhaps the authors could reanalyze their data comparing different genres/ styles?). Moreover, the stimuli were very short (7 sec), making it difficult to evoke any sort of aesthetic reactions or emotions.
- Stimuli selection procedure:
- The authors mention in line 137 that the music excerpts were selected in a balanced way in terms of several music properties (e.g., rhythm, melody, speed, dynamics, etc). This seems almost an impossible task. How did the authors achieve such balance?
- Music excerpts were selected a priori by 5 music experts. This decision is introducing an important bias in the selected stimuli, in particular, if the stimuli are then rated both by musicians and non-musicians. Perhaps if the stimuli had been previously selected by non-musicians, then the authors would have found differences in those categories.
- In Experiment 2, I worry that the lack of effects could be in part explained by the use of 2 different music sets. When listening to music, each exposure to novel stimuli is particularly unique. With repeated exposure, participants then quickly learn the statistical probabilities of the music, creating expectations and usually increasing music enjoyment with repeated exposure. By presenting two different sets of music stimuli, perhaps the authors removed any chance of increased aesthetic stimuli in the second exposure (which were all novel stimuli). An alternative paradigm would be to present the same repeated set two times. Here, we would expect an increase in liking and familiarity in the second repetition. However, this increase could be compared between a control group (no brain stimulation) and a group that received brain stimulation, expecting a higher increase in the latter. See for example the paradigm used in Aydogan et al. (2018):
Suggested references:
Müllensiefen, D., Gingras, B., Musil, J., & Stewart, L. (2014). The musicality of non-musicians: an index for assessing musical sophistication in the general population. PloS one, 9(2), e89642.
Zentner, M., Grandjean, D., & Scherer, K. (2008). Emotions evoked by the sound of music: Characterization, classification, and measurement. Emotion, 8(4), 494–521.
Aydogan, G., Flaig, N., Ravi, S. N., Large, E. W., McClure, S. M., & Margulis, E. H. (2018). Overcoming bias: Cognitive control reduces susceptibility to framing effects in evaluating musical performance. Scientific reports, 8(1), 1-9.
Author Response
Thank you for your precious suggestions and constructive comments.
- The rationale of the study and clarity: It was not clear to me what is the rationale behind the two experiments reported in this paper. Moreover, the introduction does not really help in clarifying those. In particular:
- Why did the authors expect in the first place any differences in the aesthetic responses of music by expert musicians and non-musicians? The fact that there exist differences in the auditory sensorimotor and subcortical regions in the two groups, does not necessarily mean they should have different aesthetic responses to music. The authors should clarify this in the first paragraph of the introduction when explaining the motivation of Experiment 1.
Thank you. We added specifications in lines 72-78.
- Similarly, what is the connection between Experiment 1 and Experiment 2? Experiment 1 examines the effect of music expertise on aesthetic responses to music, whereas Experiment 2 studies the role of a particular brain region involved in aesthetic appreciation. This should also be clarified in the paper.
The motivation was clarified in lines 87-89 and in lines 224-226.
- The introduction would benefit from stating clearly the hypotheses that guided this work. This would also help to improve the clarity in the results section.
Thank you, we underlined the general hypothesis at the end of the introduction (lines 118-123)
- In terms of clarity and fluidity, it would help to present first the methods and results of Experiment 1, and then do the same for Experiment 2 (potentially adding a short discussion after each experiment). Currently, presenting the methods of the two experiments in succession and then the same for the results is a bit confusing.
Thank you for your suggestion. We adopted the solution advised.
- Some information seems to be repeated, for example in the procedure of Experiment 2 (see line 49 and line 291).
The repetitions were deleted, and corrections were applied.
- Some parts of the manuscript would benefit from English proofreading.
Thank you. We asked an English mother tongue colleague to review English in our manuscript.
- Contribution and methodological limitations: Overall, it’s not clear what is the contribution of this work. Is the contribution that the null results suggest different brain networks for aesthetic appreciation in music and art. If so, this is a very strong claim based on the experiments and results reported in the paper. On the other hand, the contribution may be that the study shows that there are no differences in aesthetic appreciation depending on expertise. In this case, I’m not convinced that this is a very important contribution by itself, but perhaps I could be convinced otherwise (e.g., why would we expect differences based on expertise? Is this a common finding in other domains?). Importantly, the observed results could be explained by several limitations of the design and methods used in the experiments.
Thank you for your constructive comment. We modified our final claims both in the abstract (lines 22-24) and in the conclusions (lines 769-771), in order to focus only on music. As for the differences based on the expertise, we clarified our motivations in the introduction (lines 72-78).
Specifically:
- Measuring musicality: it seems that the authors only used “previous music training” to classify participants as non-musicians. However, musicality is a complex construct that depends on several other dimensions other than formal training. For example, a DJ or music blogger may know a lot about music without having any previous formal training. The golden standard method in music cognition to measure musical expertise, the GOLD-MSI, identifies 5 musicality dimensions: musical training, active engagement to music, perceptual abilities, singing abilities, and music emotionality. In a study where the differences between musicians and non-musicians are so important, it would help a lot to understand better how different the two groups really were in terms of their musicality. I strongly recommend to the authors using this tool in the future to measure music expertise (see reference below, Müllensiefen et al., 2014):
Thank you for the suggested tool. We will surely adopt it for our future studies. Furthermore, thank you the interesting topic you proposed that we will surely investigate in our future studies. As for the present study, we actually excluded also people with special interest in music (as explained in lines 130 and 223), therefore also DJs or music bloggers from the control group (a specification was added). On the other hand, the level of expertise of DJs or music bloggers is not necessarily comparable to the professional musicians’ one and therefore we decided to insert the criterium of the conservatory in order to certificate musicians’ skills, as suggested by previous studies (e.g., Mas-Herrero, E., Marco-Pallares, J., Lorenzo-Seva, U., Zatorre, R. J., & Rodriguez-Fornells, A. (2012). Individual differences in music reward experiences. Music Perception: An Interdisciplinary Journal, 31(2), 118-138.)
- Similarly, there are validated tools to study emotional reactions to music, which is a complex and huge area of research in music psychology and music cognition (not measuring emotions evoked by music properly is, therefore, an important problem). See for example Zentner et al. 2008 (full reference below).
Thank you for your precious suggestion. Music can indeed evoke a wide range of emotions. The GEMS is an extremely interesting and useful tool, but require a relatively long time to be completed and also a high level of emotional awareness of participants to be able to distinguish between different emotions. On the other hand, the purpose of our study was not to ask participants which emotions were evoked, but their valence: positive, negative or neutral, in order to confront the results with a similar task requiring to express an aesthetic judgment, otherwise impossible to realise. Also, focusing on the valence guaranteed us that all participants were suitable for our experiments and that the experiments required a shorter time.
- Musicians were slower than non-musicians: the authors argue that this result may be explained by the idea that musicians’ judgements were more analytical and required higher cognitive effort. However, the results could alternatively be explained by the fact that musicians enjoyed the music listening task more than non-musicians and therefore were on average slower.
Thank you for your constructive comment. We decided to justify our results referring to a more analytical judgment and a higher cognitive effort, since we found previous findings in literature which supported this hypothesis (Criscuolo, A., Pando-Naude, V., Bonetti, L., Vuust, P., & Brattico, E. (2021). Rediscovering the musician's brain: a systematic review and meta-analysis. bioRxiv; Porflitt, F., & Rosas, R. (2020). Core music elements: rhythmic, melodic and harmonic musicians show differences in cognitive performance. Studies in Psychology, 41(3), 532-562.). Secondly, following your hypothesis, we would have expected a difference also in the judgment, and not only in the response latencies. On the other hand, it will be surely interesting to investigate if a more enjoyable musical task could reduce or increase the time needed to answer in musicians and laypersons.
- Rating task and stimuli: The null effects found between the two groups could be explained by the experimental task and stimuli used. The authors did a good job in sampling the music stimuli and including a wide variety of sounds representing music. However, it is likely that if differences in aesthetic responses to music exist in the two groups, these are concentrated in particular music genres or styles (e.g., classical music for formally trained musicians or jazz music for jazz musicians). Thus, controlling for and specifically studying different music genres is an important step (perhaps the authors could reanalyse their data comparing different genres/ styles?). Moreover, the stimuli were very short (7 sec), making it difficult to evoke any sort of aesthetic reactions or emotions.
Thank you for your precious suggestion. It is not possible to reanalyse our data comparing different genres since we included a lot of different genres in order to exclude the influence of specific music genres on our results, but surely it would be very interesting to conduct a future study focused on the exploration of eventual differences between different music genres.
- Stimuli selection procedure:
- The authors mention in line 137 that the music excerpts were selected in a balanced way in terms of several music properties (e.g., rhythm, melody, speed, dynamics, etc). This seems almost an impossible task. How did the authors achieve such balance?
We added a specification in line 152.
- Music excerpts were selected a priori by 5 music experts. This decision is introducing an important bias in the selected stimuli, in particular, if the stimuli are then rated both by musicians and non-musicians. Perhaps if the stimuli had been previously selected by non-musicians, then the authors would have found differences in those categories.
Thank you for your constructive comment. We decided to select musical excerpt a priori and to ask professional musicians to categorise them in order to obtain an analytic and expert division of the stimuli in terms of rhythm, melody, etc. On the other hand, this is a very interesting point that we will surely take in consideration in our future studies.
- In Experiment 2, I worry that the lack of effects could be in part explained by the use of 2 different music sets. When listening to music, each exposure to novel stimuli is particularly unique. With repeated exposure, participants then quickly learn the statistical probabilities of the music, creating expectations and usually increasing music enjoyment with repeated exposure. By presenting two different sets of music stimuli, perhaps the authors removed any chance of increased aesthetic stimuli in the second exposure (which were all novel stimuli). An alternative paradigm would be to present the same repeated set two times. Here, we would expect an increase in liking and familiarity in the second repetition. However, this increase could be compared between a control group (no brain stimulation) and a group that received brain stimulation, expecting a higher increase in the latter. See for example the paradigm used in Aydogan et al. (2018):
Thank you for your constructive comment. We wanted to be sure to exclude an effect of increased familiarity to the musical excerpt that could possibly positively influence participants’ judgments, according to previous findings in neuroasthetics (e.g., Pereira, C. S., Teixeira, J., Figueiredo, P., Xavier, J., Castro, S. L., & Brattico, E. (2011). Music and emotions in the brain: familiarity matters. PloS one, 6(11), e27241.). Therefore, we adopted different set of musical excerpts, although superimposable, in order not to incur into misleading results
Reviewer 3 Report
The authors presented findings on the aesthetic appreciation of musical stimuli in musical experts and non-musical experts. The study applied one session tDCS on DLPFC in 20 right-handed subjects in each group. The study yielded a null result regarding tDCS judgments. The study presented behavioral findings in the sense that musical experts have a longer reaction time to the musical judgment which is commented by their analytical musical expertise.
-It is suggested to authors to present the findings related to single-session tDCS and possible effects on human behavior (different study settings can be presented) after 20 min session, which might be an argument for making the submitted study design. Present biological foundation for tDCS effects.
-Approximately how long each experiment lasts? Exp 1 and Exp 2, individually. Did the subject have the pause?
-row 475-477, please verify the sentence “The main effect of Session, F (1, 21) = 4.49, p = .046, ηp2 = .18, was significant, with lower response latencies during the pre-tDCS task (M±SD = 1427±541 msec) than during the post-tDCS task, (M±SD= 1322±518 msec).
Did the authors mean: ” “The main effect of Session, F (1, 21) = 4.49, p = .046, ηp2 = .18, was significant, with increased reaction time (response latencies) during the pre-tDCS task (M±SD = 1427±541 msec) than during the post-tDCS task, (M±SD= 1322±518 msec)”
-row 561, use tDCS acronym thought the whole manuscript after the first appearance of the full term (row 252?) The same is true for the acronym TMS and its full term please be consistent
-The manuscript is missing the limitation paragraph. The authors may comment on the fact related to single tDCS exposure and effects of tDCS in other conditions /maybe rehabilitation issues to tackle since to induce excitability changes in some conditions approximately 20 treatments might be necessary. Maybe comment on the relatively young population included in the study and the number of subjects that is relatively not high (lower than 30 in each group). May there be differences in the experience of musical taste in older experienced individuals? Maybe comment on the cortical area stimulated? May there be some influence to get the null result?
-Conclusion, row 614, please rewrite the sentence, since this study did not investigate visual art. Also, the same comment is related to the abstract and this comparison to visual art is suggested to be moved. In the abstract also references are not suggested to be used (line 21).
-check the term “naïve “ in the whole manuscript is it grammatically correct or “naive”?
-row 144. The full stop is missing “ ..interest in music The musician..”
Author Response
Thank you for your precious suggestions and constructive comments.
-It is suggested to authors to present the findings related to single-session tDCS and possible effects on human behavior (different study settings can be presented) after 20 min session, which might be an argument for making the submitted study design. Present biological foundation for tDCS effects.
Thank you, specifications on biological foundation for tDCS effects were added (lines 116-118).
-Approximately how long each experiment lasts? Exp 1 and Exp 2, individually. Did the subject have the pause?
As for Exp #1, the information further specifications were added in lines 188-189. For Exp #2, lines 305-310.
-row 475-477, please verify the sentence “The main effect of Session, F (1, 21) = 4.49, p = .046, ηp2 = .18, was significant, with lower response latencies during the pre-tDCS task (M±SD = 1427±541 msec) than during the post-tDCS task, (M±SD= 1322±518 msec).
Did the authors mean: ” “The main effect of Session, F (1, 21) = 4.49, p = .046, ηp2 = .18, was significant, with increased reaction time (response latencies) during the pre-tDCS task (M±SD = 1427±541 msec) than during the post-tDCS task, (M±SD= 1322±518 msec)”
We would like to thank the reviewer for noticing this point. We modified that sentence, accordingly.
-row 561, use tDCS acronym thought the whole manuscript after the first appearance of the full term (row 252?) The same is true for the acronym TMS and its full term please be consistent
Thank you. We corrected the sentences.
-The manuscript is missing the limitation paragraph. The authors may comment on the fact related to single tDCS exposure and effects of tDCS in other conditions /maybe rehabilitation issues to tackle since to induce excitability changes in some conditions approximately 20 treatments might be necessary. Maybe comment on the relatively young population included in the study and the number of subjects that is relatively not high (lower than 30 in each group). May there be differences in the experience of musical taste in older experienced individuals? Maybe comment on the cortical area stimulated? May there be some influence to get the null result?
Thank you, we added the considerations suggested in the Conclusions paragraph.
-Conclusion, row 614, please rewrite the sentence, since this study did not investigate visual art. Also, the same comment is related to the abstract and this comparison to visual art is suggested to be moved. In the abstract also references are not suggested to be used (line 21).
References were removed and the final claims corrected both in the Abstract and in the Conclusions paragraph.
-check the term “naïve “ in the whole manuscript is it grammatically correct or “naive”?
According to Cambdrige Dictionary, both forms are correct (https://dictionary.cambridge.org/dictionary/english/naive).
-row 144. The full stop is missing “ ..interest in music The musician..”
We would like to thank the reviewer for noticing this point, we corrected the typo.
Round 2
Reviewer 2 Report
Thank you for addressing the points raised in my review. While some of the points are addressed by providing new information in the text or some modifications, other points are not really addressed in the manuscript and the authors simply mention they will try to do so in future research. Given that the current manuscript does not have a limitations paragraph in the discussion, I would recommend writing a “limitations” section/ paragprah in the discussion specifically aimed to raise the most important limitations and provide indications for future research.
Also, I’m not convinced with the way the authors addressed the following point:
--
- The rationale of the study and clarity: It was not clear to me what is the rationale behind the two experiments reported in this paper. Moreover, the introduction does not really help in clarifying those. In particular:
- Why did the authors expect in the first place any differences in the aesthetic responses of music by expert musicians and non-musicians? The fact that there exist differences in the auditory sensorimotor and subcortical regions in the two groups, does not necessarily mean they should have different aesthetic responses to music. The authors should clarify this in the first paragraph of the introduction when explaining the motivation of Experiment 1.
Thank you. We added specifications in lines 72-78.
--
Are these two studies [i.e., 27, 28] the only justification to compare musicians with non-musicians in the current paper? The two studies do not represent the wide range of music psychology research that compares musician’s vs non-musicians’ aesthetic reactions to music. In fact, there are many studies showing no differences in aesthetic reactions between these two groups. Only to mention two:
Anglada-Tort, M., Steffens, J., & Müllensiefen, D. (2019). Names and titles matter: The impact of linguistic fluency and the affect heuristic on aesthetic and value judgements of music. Psychology of aesthetics, creativity, and the arts, 13(3), 277
Anglada-Tort, M., & Müllensiefen, D. (2017). The repeated recording illusion: the effects of extrinsic and individual difference factors on musical judgments. Music Perception: An Interdisciplinary Journal, 35(1), 94-117
Moreover, the cited papers used in text [27, 28] to justify this comparison, where “musicians preferred more country music than non-musicians”, could be explained simply by the fact that preferences for country music is higher in non-musicians than musicians in the general population. Trying to think about the best justification to compare expert and non-experts in the current paper, I wonder whether the most convincing literature is that of expert vs non-expert differential effects on judgments and decision making in various domains (e.g., there are reasons to believe that experts make different judgements that non-experts in their domain, therefore, comparing the two groups in the context of aesthetic evaluations is interesting).
Author Response
Thank you for addressing the points raised in my review. While some of the points are addressed by providing new information in the text or some modifications, other points are not really addressed in the manuscript and the authors simply mention they will try to do so in future research. Given that the current manuscript does not have a limitations paragraph in the discussion, I would recommend writing a “limitations” section/ paragraph in the discussion specifically aimed to raise the most important limitations and provide indications for future research.
Thank you for this useful suggestion. We added a limitations paragraph (lines 657-661).
Also, I’m not convinced with the way the authors addressed the following point:
--
- The rationale of the study and clarity: It was not clear to me what is the rationale behind the two experiments reported in this paper. Moreover, the introduction does not really help in clarifying those. In particular:
- Why did the authors expect in the first place any differences in the aesthetic responses of music by expert musicians and non-musicians? The fact that there exist differences in the auditory sensorimotor and subcortical regions in the two groups, does not necessarily mean they should have different aesthetic responses to music. The authors should clarify this in the first paragraph of the introduction when explaining the motivation of Experiment 1.
Thank you. We added specifications in lines 72-78.
--
Are these two studies [i.e., 27, 28] the only justification to compare musicians with non-musicians in the current paper? The two studies do not represent the wide range of music psychology research that compares musician’s vs non-musicians’ aesthetic reactions to music. In fact, there are many studies showing no differences in aesthetic reactions between these two groups. Only to mention two:
Anglada-Tort, M., Steffens, J., & Müllensiefen, D. (2019). Names and titles matter: The impact of linguistic fluency and the affect heuristic on aesthetic and value judgements of music. Psychology of aesthetics, creativity, and the arts, 13(3), 277
Anglada-Tort, M., & Müllensiefen, D. (2017). The repeated recording illusion: the effects of extrinsic and individual difference factors on musical judgments. Music Perception: An Interdisciplinary Journal, 35(1), 94-117
Moreover, the cited papers used in text [27, 28] to justify this comparison, where “musicians preferred more country music than non-musicians”, could be explained simply by the fact that preferences for country music is higher in non-musicians than musicians in the general population. Trying to think about the best justification to compare expert and non-experts in the current paper, I wonder whether the most convincing literature is that of expert vs non-expert differential effects on judgments and decision making in various domains (e.g., there are reasons to believe that experts make different judgements that non-experts in their domain, therefore, comparing the two groups in the context of aesthetic evaluations is interesting).
Thank you for raising up this point. We enriched the introduction with other references and one of the studies suggested (lines 65-67; 76-93; 135-139).
Reviewer 3 Report
Thank you for clarifying raised issues.
Author Response
Thank you for appreciating our manuscript and for your precious suggestions.